# Hybrid Space Vector PWM Strategy for Three-Phase VIENNA Rectifiers

**DOI:** 10.3390/s22176607

**Published:** 2022-09-01

**Authors:** Yaodong Wang, Yinghui Li, Xu Guo, Shun Huang

**Affiliations:** 1Aviation Engineering School, Air Force Engineering University, Xi’an 710038, China; 2Graduate School, Air Force Engineering University, Xi’an 710038, China

**Keywords:** Vienna rectifier, space vector pulse width modulation, discontinuous pulse width modulation, current distortion

## Abstract

Vienna rectifiers are widely used, but they have problems of zero-crossing current distortion and midpoint potential imbalance. In this paper, an improved hybrid modulation strategy is proposed. According to the phase difference between the reference voltage vector and the input current vector, the dynamic current crossing distortion sector is divided at each phase current crossing, and the discontinuous pulse width modulation strategy is adopted in the sector to suppress the redundant small vector mutation and improve the current crossing distortion. To solve the problem of midpoint potential balance, a space vector modulation strategy is adopted outside the sector to control the midpoint potential balance by allocating the time of small vector action by voltage balance factor. Finally, the effectiveness of the proposed method is verified by simulation and experiment.

## 1. Introduction

With the development of science and technology, the functions of industrial products have become more and more powerful, the system complexity has become higher and higher, and the correlation between systems has become closer. The nonlinear load is affected more and more, because when the nonlinear load in the system works abnormally, the power supply quality of the power supply is often affected, and the performance index of the sensor is affected by the power supply quality. Therefore, it is necessary to improve the power supply quality of the sensor power supply. The Vienna rectifier [1,2] uses a bidirectional switching structure and has lower power consumption and cost than the traditional three-phase six-switch two-level PWM rectifier. It is widely used in wind power generation systems [3,4], aircraft power supply systems and electric vehicle charging piles [5,6] due to its high power density and no dead zone driven by the power switch.

The performance of the Vienna rectifier is largely affected by the modulation strategy. The traditional discontinuous pulse width modulation strategy [7,8] (DPWM) adopts the five-stage synthesis method, which has low power consumption [9] and no zero-crossing distortion, but poor midpoint voltage balance ability [10]. The space vector modulation strategy [11,12,13] (SVPWM) adopts the seven-stage synthesis method, which has good midpoint potential regulation ability [14], but hysteresis and mutation [15] will occur when current passes zero, thus reducing power quality. The literature [16] proposed an improved spatial voltage vector modulation strategy. By calculating the relation between the midpoint potential fluctuation value and the correction amount needed to eliminate the vector error, the correct vector action time is obtained, which improves the input current quality, but only applies to the case of small DC side capacitance. The literature [17] proposed CBPWM based on dual carrier and dual modulation waves, which can be equivalent to VSVPWM. It solves the problems of complex SVPWM partition, large computation of vector action time and large power consumption. However, the input current distortion is not considered. In response to input current distortion caused by buffer circuits and parasitic capacitors in switch devices of Vienna rectifiers, the literature [18] proposes a carrier-based discontinuous space vector modulation method with variable clamping interval, which has better input current quality than conventional DPWM. However, the midpoint potential balance ability is poor. The literature [7] proposed an improved DPWM method to adjust midpoint voltage by using a redundant clamp mode. It improves the efficiency of the rectifier and reduces the power consumption of the rectifier. The literature [19] proposed a new type of hybrid modulation method, the dynamic adjustment factor combined with capacitive voltage deviation control, to improve the ability of Vienna rectifiers’ midpoint voltage balance.

In view of the above research background, combining the advantages of various modulation strategies, a hybrid modulation strategy based on current zone clamp mode is proposed, according to the phase difference between current and voltage, the current zero crossing distortion region is calculated automatically. In this sector, the DPWM modulation mode is adopted to improve the current zero crossing distortion and improve the current quality. In view of the defect that the DPWM modulation strategy cannot balance the midpoint, the SVPWM modulation strategy is adopted in the current zero crossing distortion area to make the system have better midpoint voltage regulation ability.

## 2. Vienna Rectifier Working Principle and Current Distortion Causes

Figure 1 shows the structure of the T-type three-phase Vienna rectifier. *e*_a_, *e*_b_, and *e*_c_ are the three-phase AC voltage sources. L is the filter inductance of the three-phase AC side, R is the equivalent series resistance of the three-phase AC side.

*S*_a_, *S*_b_, and *S*_c_ are bidirectional power switches connected to the neutral point on the DC side. Each bidirectional power switch consists of two MOSFET switch components and two diodes. C_1_ and C_2_ are filter capacitors on the DC side, and R_L_ is the load.

The switch component *S*_a_ is used as an example to analyze the working mode of the Vienna rectifier. *S*_x_ = 0 (x = a, b, c) indicates that the switch component *S*_x_ is disconnected, and *S*_x_ = 1 indicates that the switch component *S*_x_ is closed. *i*_x_ > 0 indicates that the current direction is from the AC side to the DC side. *i*_x_ < 0 indicates that the current direction is from the DC side to the AC side. As shown in Figure 2, when *S*_a_ = 0 and *i*_a_ > 0, the current flows through the diode to the positive terminal of capacitor C_1_, and the U_ao_ branch voltage is 1/2 Vdc. When *S*_a_ = 0 and *i*_a_ < 0, the current flows from the negative terminal of capacitor C_2_ to AC measurement, and the U_ao_ branch voltage is −1/2 Vdc. When *S*_a_ = 1, the current is connected to O point and the U_ao_ branch voltage is 0. It can be seen that the Vienna rectifier’s space vector consists of 25 basic vectors, as shown in Figure 3, including 6 large vectors, 6 medium vectors, 12 small vectors, and 1 zero vector, divided into 6 large sectors, and each large sector is divided into 6 small sectors. In the figure, the letter P represents that the bridge arm voltage of the phase is 1/2 Vdc (P level), the letter N represents that the bridge arm voltage of the phase is −1/2 Vdc (N level), and the letter O represents that the bridge arm voltage of the phase is 0 (O level).

As shown in Figure 4, *E*_d_ is the input voltage, *U*_L_ is the inductive voltage, *i*_d_ is the input current, V_ref_ is the reference voltage vector, because there is a filter inductor L on the AC side. The input current generates a voltage *U*_L_ on the filter inductor L, whose phase is 90° ahead of the input voltage E_d_, resulting in a phase difference θ between the reference voltage V_ref_ and the input current *i*_d_. When V_ref_ does not pass through zero of phase b, the current *i*_d_ has passed through zero and the sign changes from negative to positive. According to the working principle of Vienna rectifiers, at *S*_b_ = 0 the bridge arm voltage is positive or negative depending on the direction of the current. Therefore, the vector [ONN] is replaced by [OPN], as shown in Figure 5 in actual work, resulting in the wrong trigger signal and thus distortion of input current.

According to the phase relationship in Figure 4, the θ expression can be obtained, where ω is the angular frequency of the AC power supply.
(1)$$\theta =\arctan\frac{U_{L}}{E_{d}}=\arctan\frac{\omega Li_{d}}{E_{d}}$$

There are six zero-crossing distortion regions in one cycle, and the distribution is shown in Figure 6. The replacement of redundant vectors is shown in Table 1.

## 3. Improved Hybrid Modulation Strategy

According to the previous analysis of the causes of current zero-crossing distortion, it can be known that to overcome the distortion of the current near the zero-crossing point, it is necessary to ensure that any phase current changes from positive to negative or from negative to positive without causing sudden changes of redundant small vectors (such as [PPO] [OON]) that appear in pairs. Therefore, the corresponding phase voltage vector is clamped to zero when the current crosses zero.

The discontinuous pulse width modulation strategy (DPWM) meets the above-mentioned requirements that the current zero-crossing does not cause a sudden change of the redundant small vector, but there are problems that the current ripple is large and the midpoint potential cannot be adjusted. The space vector modulation strategy (SVPWM) has a small current ripple and a good ability to adjust the midpoint potential, but the disadvantage is that the current will be distorted when it crosses zero.

To sum up, according to the complementary advantages and disadvantages of these two modulation methods, this paper proposes a hybrid modulation strategy. First, the system detects the phase difference θ between the reference voltage and the input current and designs a zero-crossing distortion sector with an angle slightly larger than θ according to the phase difference, as shown in Figure 7. In the zero-crossing distortion sector, the Vienna rectifier adopts a discontinuous pulse width modulation strategy to solve the problem of current zero-crossing distortion. The space vector modulation strategy is adopted in the non-zero-crossing distortion sector, and the midpoint potential is adjusted by dynamically distributing the time of the small vector action by the balance factor, which ensures the balance of the midpoint potential.

Figure 8 shows the operation of the reference voltage vector in sectors 3, 4, 5, and 6. When the reference voltage is in the distortion region I, the five-stage modulation mode is adopted, and the seven-stage modulation mode is adopted in other regions. The modulation mode, clamping mode, and corresponding switch sequence of different sectors are shown in Table 2.

Figure 9 shows the working conditions of the reference voltage vector in sectors 1 and 2. When the reference voltage is in the distortion region I, a five-segment modulation method is adopted, and a seven-segment modulation method is adopted in other regions. The modulation modes, clamping modes, and corresponding switching sequences of different sectors are shown in Table 3.

Midpoint potential balance control is one of the important conditions for the Vienna rectifier to work stably. The midpoint potential of the Vienna rectifier is controlled by the charge and discharge of the capacitor from the current flowing through the two midpoints of the output capacitor. The basic vectors of the Vienna rectifier are divided into large, medium, small, and zero vectors according to the different switching modes and current directions described in the previous section.

When the large vector acts, the three-phase currents are not directly connected to the midpoint O, but flow through the upper busbar capacitor C_1_ and the lower busbar capacitor C_2_, and the midpoint potential is not affected, so the large vector does not affect the midpoint potential.

When the zero vector acts, the three-phase input currents are all connected to point O. According to Kirchhoff’s current law: at any node in the circuit, at any time, the sum of the currents flowing into the node is equal to the sum of the currents flowing out of the node. That is *i*_a_ + *i*_b_ + *i*_c_ = 0. Therefore, the zero vector does not affect the midpoint potential, either.

When the mid-vector acts, the midpoint O of the two capacitors is connected to one-phase current in the three-phase current, so the midpoint potential depends on the flow direction of the connected phase current. If the current is positive, the midpoint potential rises, and if the current direction is negative, the midpoint potential falls.

Since small vectors exist in pairs, they are divided into positive small vectors (such as [P O O][P P O]) and negative small vectors (such as [O N N][O O N]). When the positive small vector acts, the current flows from the positive terminal of the capacitor C_1_ and flows back to the AC measurement from the O point, resulting in a drop in the midpoint potential. When the negative small vector acts, the current flows in from the midpoint O of the two capacitors and flows out from the negative terminal of the capacitor C_2_, resulting in an increase in the midpoint potential.

To sum up, the large vector and the zero vector have no influence on the midpoint potential, while the medium vector and the small vector can influence the midpoint potential. Among them, the influence of the medium vector on the midpoint potential depends on the direction of the current and cannot be controlled. Therefore, the equilibrium of the midpoint potential can be controlled by choosing the time of action of the small vector.

Assume that the voltage difference between the upper and lower capacitors is Δ*U*, and *k* is the voltage balance factor. KP is the proportional coefficient, and KI is the integral coefficient.
(2)$$\Delta U=U_{c1}-U_{c2}$$
(3)k=KP×ΔU+KI×ΔU   k∈[−1,1]
(4)$$Tsz=0.5\left(1+k\right)Ts$$
(5)$$Tsf=0.5\left(1-k\right)Ts$$

Taking the first large sector as an example, the modulation timing diagram after adding midpoint balance control to SVPWM is shown in Figure 10. In Figure 10, the horizontal axis is the time of vector action, and the vertical axis is the voltage vector condition of each small sector in sector I. When the midpoint voltage is high, that is, when the voltage difference between the positive and negative capacitors on the DC side is ΔUdc > 0, the balancing loop regulator outputs *k* > 0. At this time, the action time of the positive small vector becomes longer, and the action time of the negative small vector becomes shorter, which reduces the midpoint voltage. In the same way, when the midpoint voltage is low, the output of the balancing loop regulator is *k* < 0. The action time of the negative small vector becomes longer, and the action time of the positive small vector becomes shorter, which increases the midpoint voltage.

## 4. Discussion

To verify the effectiveness of the proposed hybrid modulation strategy, a simulation model of the 5 kW Vienna rectifier was built by MATLAB/SIMULINK, and the three-phase input current, line-to-neutral voltage, and midpoint potential difference were simulated. In order to verify the performance of the improved hybrid modulation strategy proposed in this paper under different modulation ratios and different output voltages, simulations were carried out for the modulation ratio m = 0.51, the output voltage Vdc = 400 V, m = 0.86, Vdc = 600 V. Simulation parameters are shown in Table 4.

Figure 11 shows the simulation waveform when the modulation ratio m = 0.51 and the output voltage Vdc = 400 V using the traditional modulation method. From the three-phase input current i_abc_ waveform diagram, it can be seen that the harmonic distortion is more serious at the zero-crossing of each phase current, and the total harmonic distortion rate reaches 4.15%. From the line-to-neutral voltage Vao waveform, it can be seen that there is a “voltage jump” at the zero-crossing of the a-phase current, and the overall line-to-neutral voltage has a certain amplitude fluctuation. It can be seen from the midpoint potential waveform that the voltage difference between the upper and lower capacitors at the zero-crossing of the a-phase current reaches the maximum 1.2 V, and the midpoint potential difference fluctuates greatly.

Figure 12 shows the simulation waveform when the modulation ratio m = 0.51 and the output voltage Vdc = 400 V using the improved hybrid modulation method. From the three-phase input current i_abc_ waveform diagram, it can be seen that the harmonic distortion has been greatly improved at the zero-crossing of each phase current, and the total harmonic distortion rate is only 1.39%. From the line-to-neutral voltage Vao waveform, it can be seen that the positive and negative voltage switching is smooth and fast at the zero-crossing of the a-phase current, and the problem of the voltage jumping up and down is solved. It can be seen from the midpoint potential waveform that the fluctuation of the midpoint potential difference is small, and the voltage difference between the upper and lower capacitors is 0.2 V.

Figure 13 shows the simulation waveform when the modulation ratio m = 0.86 and the output voltage Vdc = 600 V using the traditional modulation strategy. From the three-phase input current i_abc_ waveform diagram, it can be seen that there is harmonic distortion at the zero-crossing of each phase current, and the total harmonic distortion rate reaches 3.86%. From the Vao waveform of the line-to-neutral voltage, it can be seen that there is a certain amplitude fluctuation in the a-phase voltage. From the midpoint potential waveform, it can be seen that the midpoint potential difference fluctuates greatly, and the voltage difference between the upper and lower capacitors at the zero-crossing of the a-phase current reaches 1.38 V.

Figure 14 shows the simulation waveform when the modulation ratio m = 0.86 and the output voltage Vdc = 600 V using the improved hybrid modulation method. From the three-phase input current i_abc_ waveform diagram, it can be seen that the harmonic distortion at the zero-crossing of the current is small, and the total harmonic distortion rate is only 1.74%. From the line-to-neutral voltage Vao waveform, it can be seen that the positive and negative voltage switching is smooth and fast at the zero-crossing of the a-phase current, and there is no problem of voltage jumping up and down. From the midpoint potential waveform, it can be seen that the fluctuation of the midpoint potential difference is small, and the voltage difference between the upper and lower capacitors is 0.15 V.

In order to further verify the correctness and practicability of the proposed improved modulation strategy, a 5 kW Vienna rectifier prototype was built and verified by experiments. The DSP chip is TMS320F28021, the switching device MOSFET is SPW47N60C3, and the diode is RHRG30120. The three-phase grid voltage was provided by a 9 kVA voltage regulator. The parameters used in the experiment were consistent with those in the simulation.

Figure 15 shows the experimental waveforms of the input current under different modulation strategies. Figure 15a is the experimental waveform of the input current of the traditional modulation strategy. It can be seen that the quality of the input current waveform is poor, and the harmonic distortion is more serious at the zero-crossing of the current, with THD = 4.52%. Figure 15b is the experimental waveform of the input current for the improved hybrid modulation strategy. As can be seen from the figure, compared with the traditional SVPWM modulation strategy, the current waveform has a higher sine degree, and the current zero-crossing harmonic distortion problem is greatly improved, with THD = 2.28%. In order to verify the performance of the modulation strategy proposed in this paper under the conditions of different modulation ratios and different output voltages, the input current waveforms were collected when the modulation ratio m = 0.86 and Vdc = 600 V. Figure 15c is the experimental waveform of the input current of the traditional modulation strategy. It can be seen that the input current waveform has obvious harmonic distortion, and the current sine is poor, THD = 4.06%. It can be seen from Figure 15d that the sine of the input current waveform of the improved hybrid modulation strategy is significantly better than that of the traditional SVPWM modulation method, and there is no current zero-crossing harmonic distortion, THD = 2.21%. In conclusion, the proposed method achieves the purpose of improving the current quality, and its effectiveness and universality are also verified.

Figure 16a,c are the line-to-neutral voltage waveforms under different modulation ratios and output DC voltages of the SVPWM modulation strategy. It can be seen that in actual work, the traditional SVPWM line-to-neutral voltage fluctuates greatly, and the voltage is unstable, which greatly interferes with the performance of the Vienna rectifier. In contrast, Figure 16b,d adopt the improved hybrid modulation strategy, the line-to-neutral voltage waveform fluctuation is small, and the stability is effectively improved. It is verified that the proposed improved modulation strategy has better line-to-neutral voltage stability performance.

Figure 17a,c are the experimental waveforms of the midpoint potential difference of the traditional modulation strategy under two modulation ratios and output voltage conditions. It can be seen that the voltage fluctuation on the upper and lower two capacitors of the traditional modulation strategy is relatively large. The neutral point potential balance is an important condition for the stable operation of the Vienna rectifier. When the voltage fluctuation of the two capacitors is too large, the Vienna rectifier will fail and stop working. Figure 17b,d are the experimental waveforms of the midpoint potential difference using the improved hybrid modulation strategy. It can be seen from the figure that in the case of different modulation ratios and output voltages, the proposed improved hybrid modulation strategy can ensure that the midpoint potentials of the two capacitors at the output of the Vienna rectifier are balanced and the fluctuations are small, with good stability.

## 5. Conclusions

According to the working principle of the Vienna rectifier, this paper analysed the causes of current zero distortion, according to the DPWM modulation strategy and the advantages and disadvantages of SVPWM modulation strategies, and a neutral voltage balance factor is proposed as a hybrid modulation strategy for improvement. According to the phase difference between the reference voltage vector and the input current, the zero-distortion region of the overcurrent is divided. In this region, DPWM modulation is adopted to avoid the change of redundant small vectors caused by the phase difference and improve the zero-crossing distortion of the current. In other regions, SVPWM modulation is used to adjust the action time of small vectors in real time by the designed voltage balance factor, which improves the midpoint potential imbalance and improves the current quality. Finally, the effectiveness of the proposed strategy is verified by simulation and experiment.

## Figures and Tables

**Figure 1 sensors-22-06607-f001:**
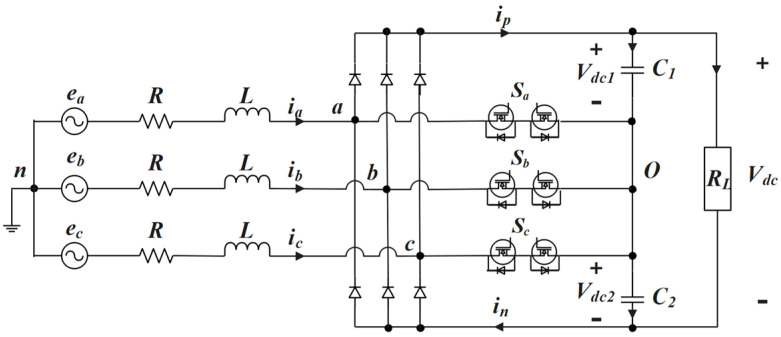
T type Vienna rectifier topology [20].

**Figure 2 sensors-22-06607-f002:**
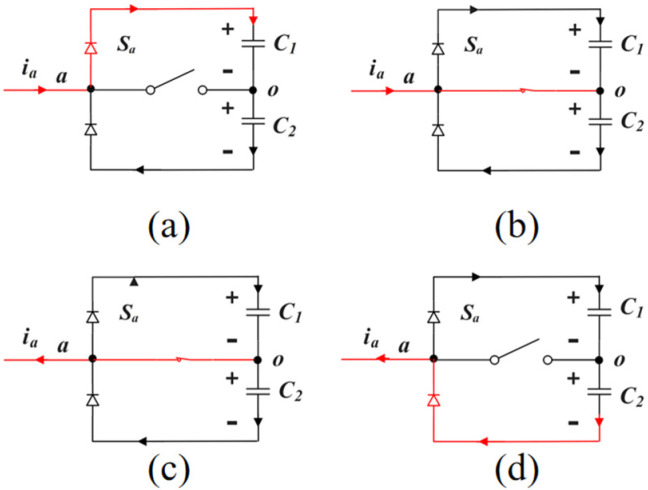
Four working states of Vienna rectifier. (**a**) *S*_a_ = 0, *i*_a_ > 0; (**b**) *S*_a_ = 0, *i*_a_ > 0; (**c**) *S*_a_ = 0, *i*_a_ > 0; (**d**) *S*_a_ = 0, *i*_a_ > 0.

**Figure 3 sensors-22-06607-f003:**
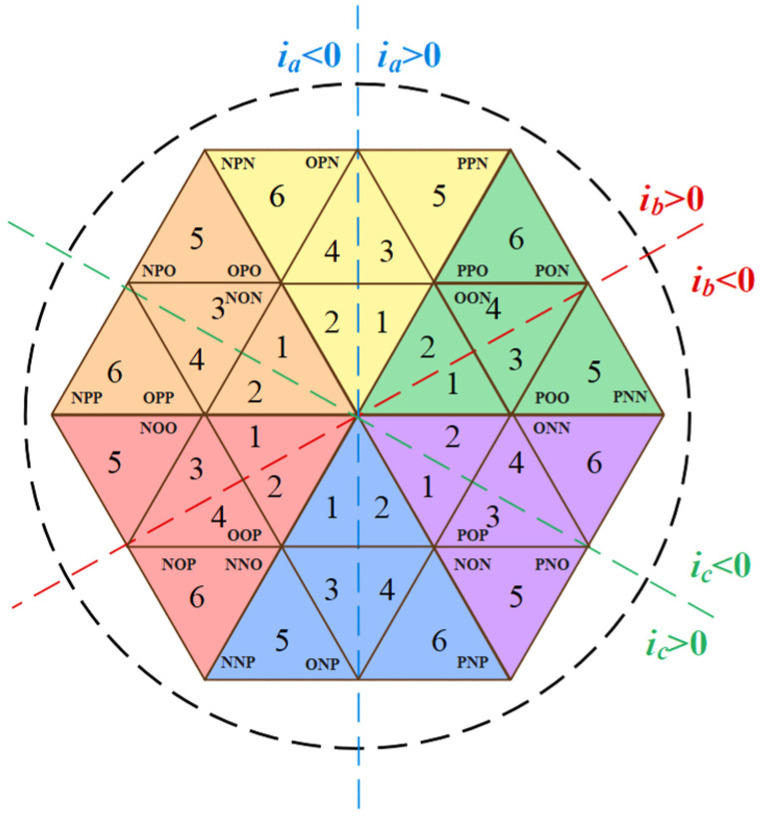
Space vector diagram of Vienna rectifier.

**Figure 4 sensors-22-06607-f004:**
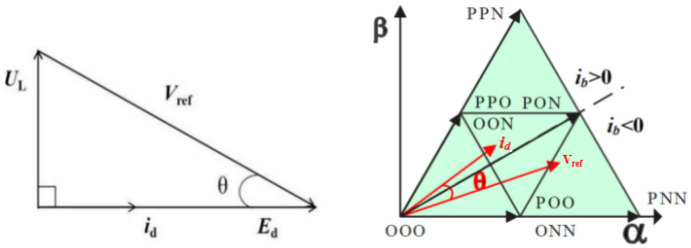
Vector relation of input voltage and current and vector diagram of large sector I space.

**Figure 5 sensors-22-06607-f005:**
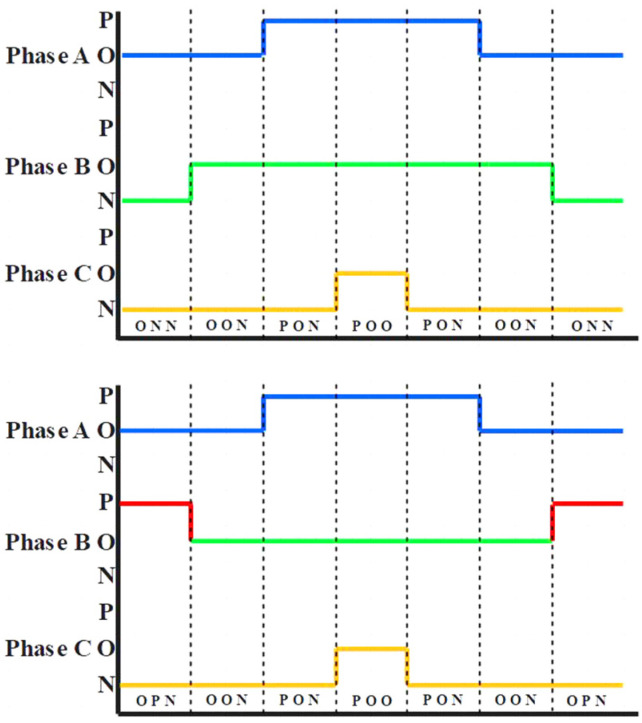
Ideal and actual voltage vector comparison.

**Figure 6 sensors-22-06607-f006:**
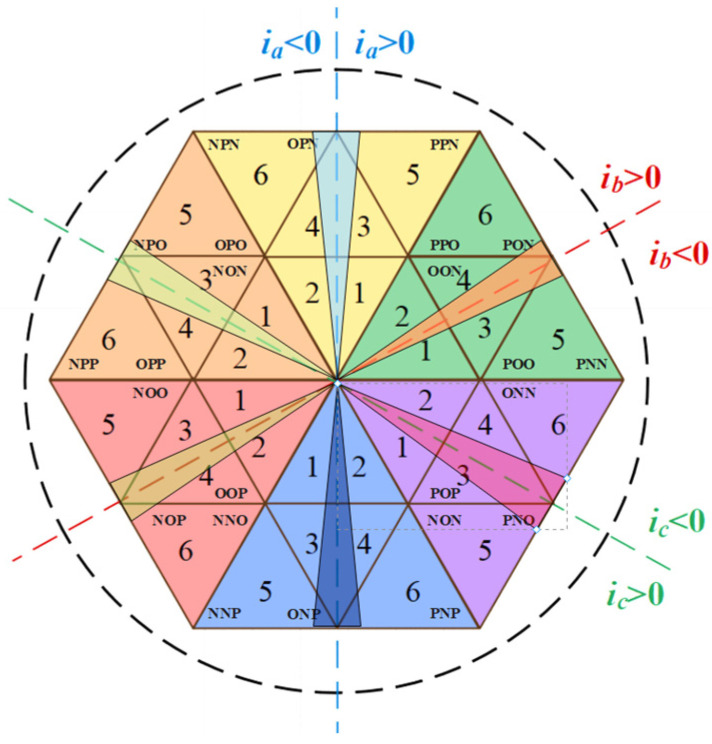
Vienna rectifier Zero-Crossing Distortion Sector Diagram.

**Figure 7 sensors-22-06607-f007:**
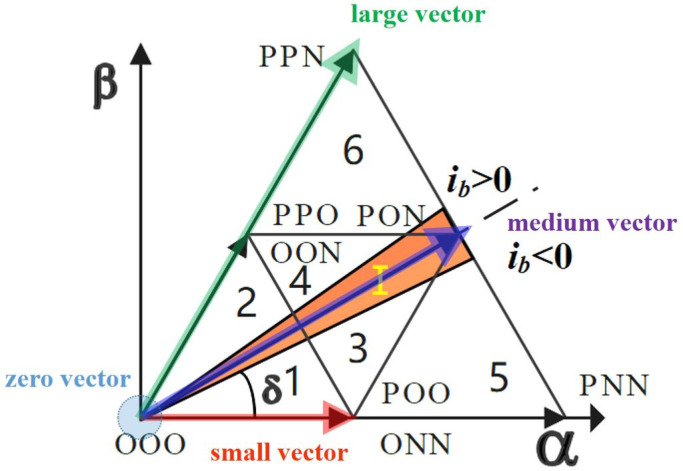
Zero-crossing distortion region in large sector I.

**Figure 8 sensors-22-06607-f008:**
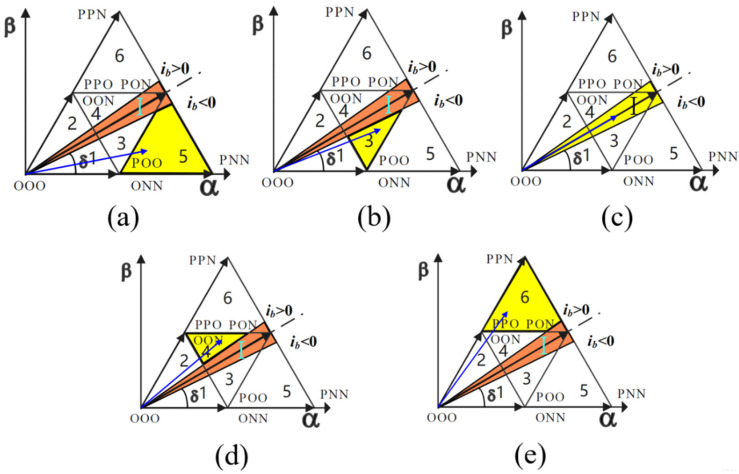
Operation of reference voltage vector in sectors 3, 4, 5, and 6. (**a**) The reference voltage vector is in the undistorted region of sector 5. (**b**) The reference voltage vector is in the undistorted region of sector 3. (**c**) The reference voltage vector is in the distortion region I. (**d**) The reference voltage vector is in the undistorted region of sector 4. (**e**) The reference voltage vector is in the undistorted region of sector 6.

**Figure 9 sensors-22-06607-f009:**
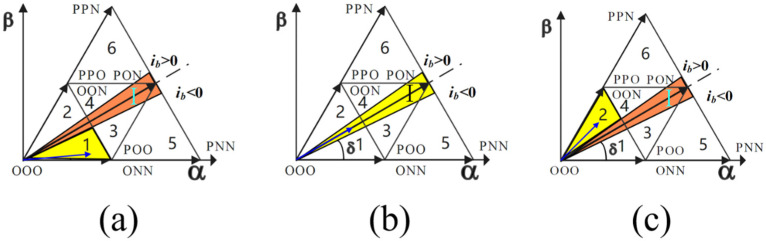
Operation of reference voltage vector in sectors 1 and 2. (**a**) The reference voltage vector is in the undistorted region of sector 1. (**b**) The reference voltage vector is in the distortion region I. (**c**) The reference voltage vector is in the undistorted region of sector 2.

**Figure 10 sensors-22-06607-f010:**
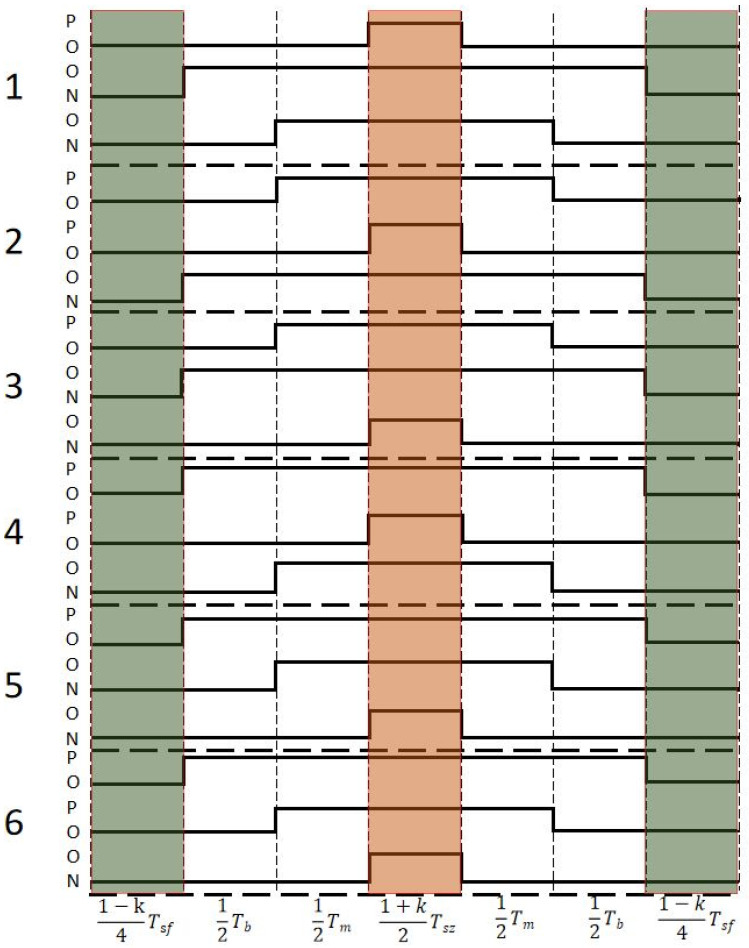
Modulation timing diagram of SVPWM after adding midpoint balance control.

**Figure 11 sensors-22-06607-f011:**
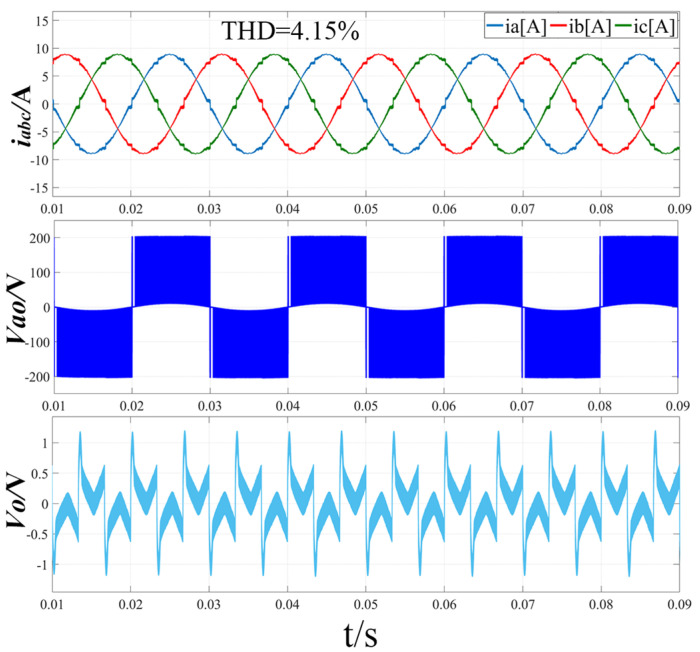
Simulation results of three-phase input current, line-to-neutral voltage and midpoint potential difference with traditional modulation strategy (m = 0.51, Vdc = 400 V).

**Figure 12 sensors-22-06607-f012:**
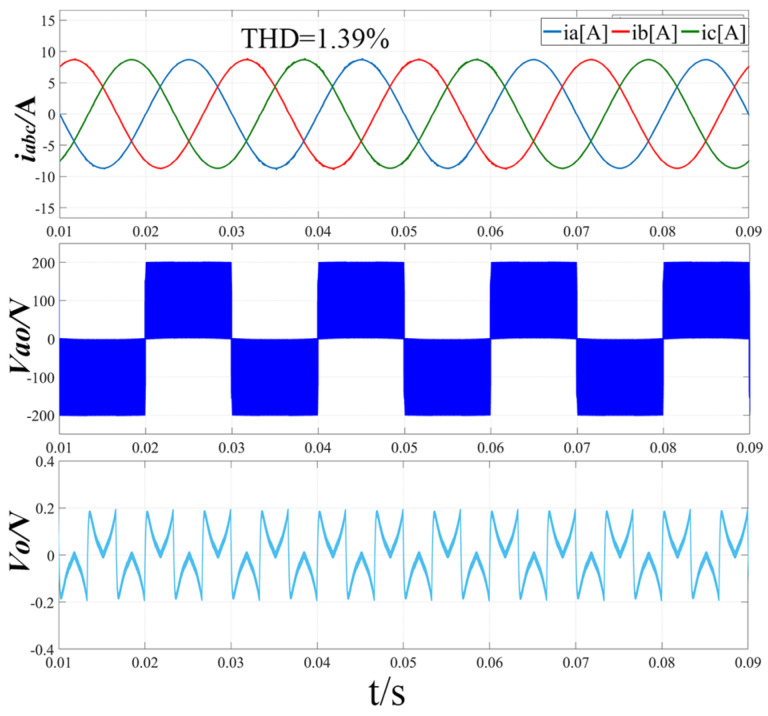
Simulation results of three-phase input current, line-to-neutral voltage and midpoint potential difference with improved hybrid modulation strategy (m = 0.51, Vdc = 400 V).

**Figure 13 sensors-22-06607-f013:**
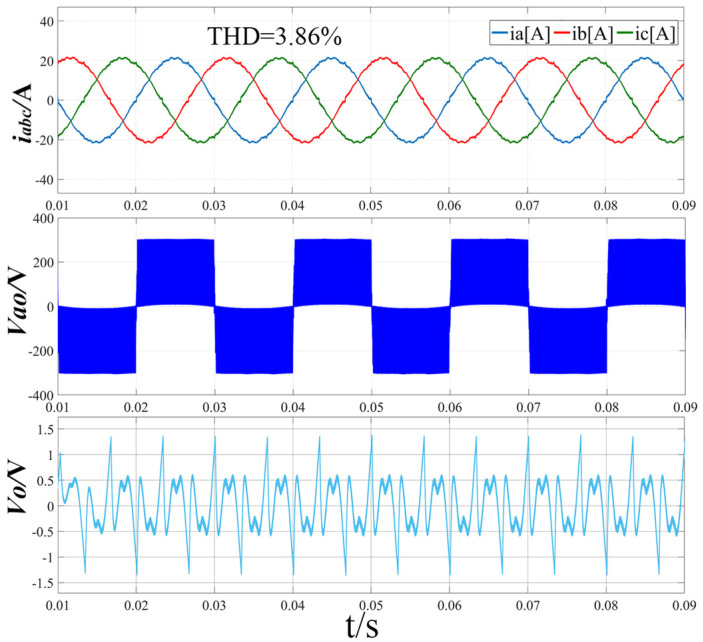
Simulation results of three-phase input current, line-to-neutral voltage and midpoint potential difference with traditional modulation strategy (m = 0.86, Vdc = 600 V).

**Figure 14 sensors-22-06607-f014:**
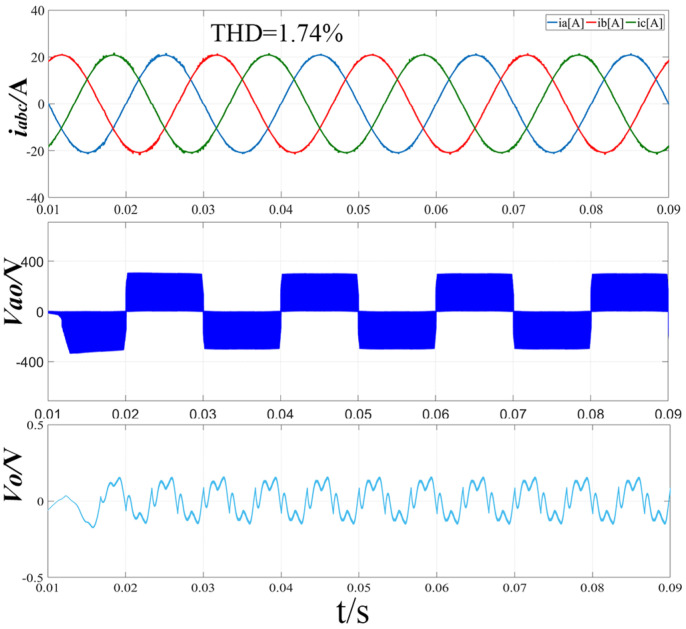
Simulation results of three-phase input current, line-to-neutral voltage and midpoint potential difference with improved hybrid modulation strategy (m = 0.86, Vdc = 600 V).

**Figure 15 sensors-22-06607-f015:**
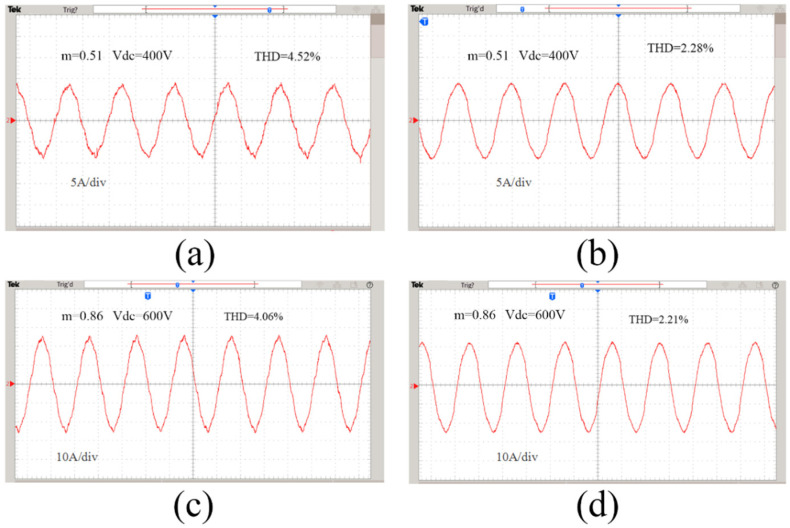
Input current experimental waveforms: (**a**) traditional method, m = 0.51, Vdc = 400 V; (**b**) proposed method, m = 0.51, Vdc = 400 V; (**c**) traditional method, m = 0.86, Vdc = 600 V; (**d**) proposed method, m = 0.86, Vdc = 600 V.

**Figure 16 sensors-22-06607-f016:**
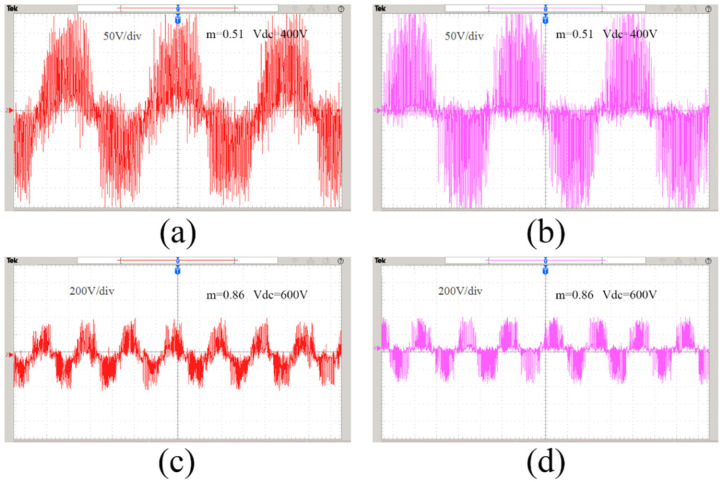
Line-to-neutral voltage experimental waveforms: (**a**) traditional method, m = 0.51, Vdc = 400 V; (**b**) proposed method, m = 0.51, Vdc = 400 V; (**c**) traditional method, m = 0.86, Vdc = 600 V; (**d**) proposed method, m = 0.86, Vdc = 600 V.

**Figure 17 sensors-22-06607-f017:**
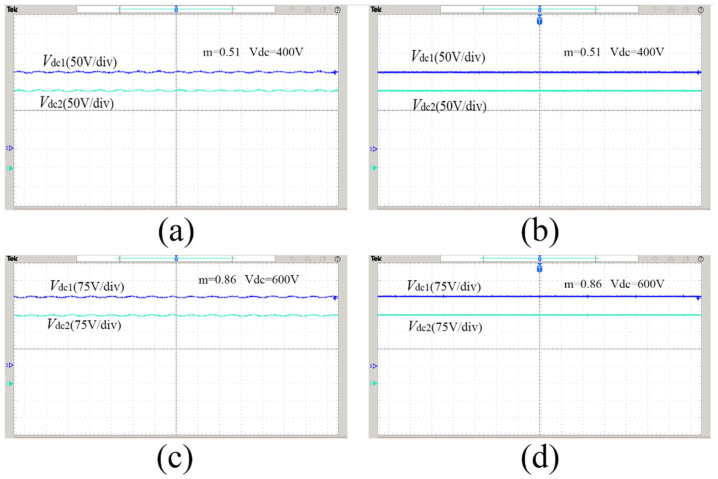
Midpoint potential difference experimental waveforms: (**a**) traditional method, m = 0.51, Vdc = 400 V; (**b**) proposed method, m = 0.51, Vdc = 400 V; (**c**) traditional method, m = 0.86, Vdc = 600 V; (**d**) proposed method, m = 0.86, Vdc = 600 V.

**Table 1 sensors-22-06607-t001:** Redundant Vector Replacement Table.

Distortion Area	Zero-Crossing Current	Ideal Vector	Actual Vector
I	*i*_b_ < 0 → *i*_b_ > 0	[O N N]	[O P N]
II	*i*_a_ > 0 → *i*_a_ < 0	[P P O]	[N P O]
III	*i*_c_ < 0 → *i*_c_ > 0	[N O N]	[N O P]
IV	*i*_b_ > 0 → *i*_b_ < 0	[O P P]	[O N P]
V	*i*_a_ < 0 → *i*_a_ > 0	[N N O]	[P N O]
VI	*i*_c_ > 0 → *i*_c_ < 0	[P O P]	[P O N]

**Table 2 sensors-22-06607-t002:** Modulation method, clamping method, and corresponding switching sequence of different sectors.

Sector	Clamp Method	Switch Sequence
5	-	[ONN] [PNN] [PON] [POO] [PON] [PNN] [ONN]
3	-	[ONN] [OON] [PON] [POO] [PON] [OON] [ONN]
I	B-O	[POO] [PON] [OON] [PON] [POO]
4	-	[OON] [PON] [POO] [PPO] [POO] [PON] [OON]
6	-	[OON] [PON] [PPN] [PPO] [PPN] [PON] [ONN]

**Table 3 sensors-22-06607-t003:** Modulation method, clamping method and corresponding switching sequence of sectors 1 and 2.

Sector	Clamp Method	Switch Sequence
1	-	[ONN] [OON] [OOO] [POO] [OOO] [OON] [ONN]
I	B-O	[POO] [PON] [OON] [PON] [POO]
2	-	[OON] [OOO] [POO] [PPO] [POO] [OOO] [OON]

**Table 4 sensors-22-06607-t004:** Key parameters of the simulation.

Parameter	Value
Three-phase voltage RMS /V	AC 115
Grid frequency/Hz	50
Switching frequency/kHz	20
Filter inductance/mH	2
DC-link capacitor/μF	2000
KP	4
KI	0.3

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
