# Peer review of "Hybrid Space Vector PWM Strategy for Three-Phase VIENNA Rectifiers"

_sensors, 2022, doi:10.3390/s22176607_

Round 1

Reviewer 1 Report

In the opinion of this reviewer, the presented article entitled  “Hybrid Space Vector PWM Strategy for Three-Phase VIENNA Rectifiers” concerns an interesting issue related to the operation of power converters. It may arouse the interest of specialists in the field.

In the opinion of this reviewer, there are fragments of the text in the article that reduce the quality of the presentation and require improvement / additions / explanations.

Some of this reviewer's suggestions are presented below:

1) Line about 25: What in this context means: abnormal work of nonlinear load.

2) Line about 28: “Vienna Rectifier[1,2] the Vienna rectifier uses a …”. This statement seems to be incorrect.

3) This statement needs explanation with respect to literature [15] cited. This statement does not seem to be supported by the article [15].

4) Caption to Fig. 1: The shown rectifier structure is different from the classic configuration shown in 1993, but is known long before the publication of the article [21] cited here.

5) Line about 51, description to Fig. 1: If this "input side" is considered a mains, shouldn't there be inductance as well?

6) Line about 71: The word "tube" seems to be inappropriate in the phrase "MOSFET switch tubes".

7) Line about 80: The word "end" seems to be inappropriate in the phrase “positive end of capacitor C1”.

8) Line about 80: It seems that the term "Uao branch voltage" instead of "bridge arm voltage Uao" would sound better here.

9) Line about 98: Id and i currents are the same currents?

10) Line about 103: It has been assumed that the term "distorted" in relation to the current waveform refers to the distortion of its shape in relation to a certain reference shape - usually sinusoidal - resulting from non-linearity or non-stationarity of circuit elements. So is the use of the term "distortion of input current" appropriate here (does it fall within the above-mentioned categories)?

11) Line about 103: What is the relation between i, id and ix (ib here?) currents?

12) Line about 140: There is no such region (i.e. region i) in Fig. 8.

13) Lines about 169-173: Two problems with the statements in lines 169-172: 1) what in a case of source-side circuit asymmetry? 2) for the analyzed circuit (shown in Fig. 1) the sum of the phase currents ia, ib and ic can be different from zero (for any method of the rectifier control method)?

15) Lines about 185-190: General question for this summing up fragment (lines 185-190): are the capacities C1 and C2 considered as exactly equal or can they differ each other?

16) Line about 192: What are the KP and KI coefficients (of what devices)?

17) Caption of Fig. 11. The term "phase voltage" as used here seems confusing. Usually, it concerns the sinusoidal phase voltage of the supply network. There is an incompatible use of this notion here - it should be adjusted/changed for better accuracy of the argument.

18) Lines about 224-225: The same problem as in 17) - see the review remark 17) to caption of Fig.11.

Summarizing the above comments, it can be stated that in order to improve the uniformity of the article, appropriate clarifications should also be made in the summary.

In the opinion of this reviewer, the presented article should be corrected in line with the suggestions presented in the review.

Reviewer 2 Report

This article describes the control of a three-phase Vienna rectifier using the hybrid space vector method PWM strategy. At the beginning of the article, the authors introduce in an interesting and substantive way the subject of control with the use of a space vector PWM strategy for a three-phase Vienna rectifier. Then they present the hybrid modulation strategy with a balancing loop regulator to reduce the voltage fluctuations on the capacitors at the bridge output and zero-crossing of the phase current.

This method has been verified both by simulation in the Matlab/Simulink and by the experiment (hardware). The results are positive and promising.

Remarks:

The authors do not provide much information about the equipment itself. What transistor drivers did they use (dead band)? Was the SVPWM algorithm calculated in real time for a frequency of 20kHz etc?

How do the voltage fluctuations of two capacitors behave in a longer time perspective, especially when the load changes?

Why the authors did not publish this article in Energies, as the topic is more relevant to this article?

Detailed Notes:

1. Strange markings on the X-axis is this a standard in "Sensors"? Usually, for example, iabc = f (t) and markings on the axes iabc [A].

2. Figures 15 and 16 do not show the scale.
